ecology/evolution

Cayo Santiago, hurricanes, matrix population models, rhesus macaque, life table response experiment

**Author for correspondence:**
Raisa Hernández-Pacheco
e-mail: rai.hernandezpacheco@csulb.edu

# Hurricane-induced demographic changes in a non-human primate population

Dana O. Morcillo[1], Ulrich K. Steiner[2],
Kristine L. Grayson[1], Angelina V. Ruiz-Lambides[3]
and Raisa Hernández-Pacheco[4]

[1]Department of Biology, University of Richmond, Richmond, VA, USA
[2]Center for Research and Interdisciplinary, Paris, France
[3]Caribbean Primate Research Center, University of Puerto Rico, Medical Sciences Campus, San Juan, Puerto Rico
[4]California State University, Long Beach, CA, USA

UKS, 0000-0002-1778-5989; KLG, 0000-0003-1710-0457;
RH-P, 0000-0002-3681-5127

Major disturbance events can have large impacts on the demography and dynamics of animal populations. Hurricanes are one example of an extreme climatic event, predicted to increase in frequency due to climate change, and thus expected to be a considerable threat to population viability. However, little is understood about the underlying demographic mechanisms shaping population response following these extreme disturbances. Here, we analyse 45 years of the most comprehensive free-ranging non-human primate demographic dataset to determine the effects of major hurricanes on the variability and maintenance of long-term population fitness. For this, we use individual-level data to build matrix population models and perform perturbation analyses. Despite reductions in population growth rate mediated through reduced fertility, our study reveals a demographic buffering during hurricane years. As long as survival does not decrease, our study shows that hurricanes do not result in detrimental effects at the population level, demonstrating the unbalanced contribution of survival and fertility to population fitness in long-lived animal populations.

## 1. Introduction

Major hurricanes are a type of extreme climatic disturbance expected to increase in frequency due to climate change [1] and thus reduce animal population viability through direct

mortality [2], habitat alteration [3], decreased food availability [4] and increased physiological stress among members of the population [5]. Although hurricanes are expected to have significant impacts on population dynamics, structure, and thus viability across animal taxa, studies on natural populations are often limited to reports on general effects among populations (e.g. total mortality) and usually do not consider within-population variation across life-history stages (e.g. age- or stage-specific mortality). Moreover, few studies examine the demographic mechanisms that determine population fitness (i.e. population growth rate) following these events during recovery. To advance our knowledge on the future dynamics and viability of animal populations following extreme climatic disturbances, we need to improve our understanding of the underlying demographic mechanisms shaping the population response across life-history stages [6,7].

Here, we aim to understand how major hurricanes affect a non-human primate population by decomposing the effect of each vital rate (survival and reproduction) on population growth rate during hurricane years. Such an acute disturbance can affect a population through decreased survival of vulnerable stages (e.g. developmental stages) [8], decreased survival of adults from increased competition [9], or reduced reproductive output due to potential trade-offs between survival and reproduction in which females allocate more energy to growth or maintenance processes, in order to ensure future reproductive potential [10]. The direct mortality of adult females is a consequence of particular interest among long-lived mammals given the significant contribution to population growth often observed from adult female survival [11,12]. The extreme nature of major hurricanes may also lead to both significant mortality and suppression of fertility across stages, raising questions regarding the adaptive potential of populations and whether impacts to viability place a population under threat of local extinction.

In this study, we analyse the long-term population dynamics of the free-ranging Cayo Santiago rhesus macaque (*Macaca mulatta*) population from 1973 to 2018 in order to examine the impacts on demography from major hurricanes during this time period. We examined years where the population experienced the direct effect of a major hurricane (category greater than or equal to 3) and use data on 4635 females structured into development and reproductive stages to parametrize matrix population models and robustly quantify variation in long-term population fitness following hurricane events. We also assess the population-level effects of variation in vital rates during hurricane years in order to determine which life cycle transition contributed the most to changes in the long-term population fitness. The Cayo Santiago rhesus macaque population is ideal for addressing demographic effects of hurricanes, given the long-term data available and its location in the Caribbean region, which has experienced the direct effects of three major hurricanes since the 1950s and is expected to experience a higher frequency of major hurricanes in the following decades due to climate change [1]. In particular, Cayo Santiago rhesus macaques bring the unique opportunity to robustly decompose the demographic effects of hurricanes into relevant life-history traits (stage-specific survival and reproduction) and allow us to know how long-lived animal populations, especially those with no long-term food limitation either due to broad diets, rapid food replenishment following a hurricane or whose principal food source is little affected by hurricanes, might respond to these extreme events.

We expected vital rates to significantly respond to the hurricanes showing stronger negative effects in fertility relative to survival due to the low temporal variability in survival reported in long-lived mammals. Our analysis reveals a decrease in mean population growth rate associated with a decreased fertility during years experiencing the impact of hurricanes; however, the population continued to grow annually due to steady survival. In this way, our study demonstrates the unbalanced contribution of survival and fertility to the long-term fitness of the population in long-lived animals.

# 2. Methods

## 2.1. The Cayo Santiago field station

Cayo Santiago is a 15.2 ha island located 1 km off the southeastern coast of Puerto Rico (lat. 18°09′ N, long. 65°44′ W) managed by the Caribbean Primate Research Center (CPRC) of the University of Puerto Rico. The Cayo Santiago Field Station (CSFS) serves as a research site for behavioural and non-invasive research on free-ranging rhesus macaques (*Macaca mulatta*). For this purpose, the rhesus macaque population was established in 1938 from 409 Indian monkeys and no other individuals have been introduced since then. Since establishment, the population has been maintained under

semi-natural conditions allowing for the natural formation of social groups, social rank and annual birth seasons (more than 72% of births occurring within a period of three months [13]). Monkeys forage on vegetation and have free access to both feeding and watering stations. They are provisioned with monkey chow at a *per capita* basis of approximately 0.23 kg/animal/day; however, 50% of their feeding time is spent on natural vegetation [14]. More importantly, individuals have social rank-related differential access to commercial feed and for those that do, the amount consumed varies significantly [15]. Until 20 September 2017, food was distributed among three open feeding stations located at different sites on the island and later was distributed along open spaces in trails after damage from Hurricane Maria. Rainwater is collected in catchments on the island, stored in concrete or fibreglass cisterns, and chlorinated prior to distribution at automatic watering stations.

Veterinary intervention is restricted to the annual trapping season where yearlings are captured, marked for identification using ear notches and a unique three-character ID tattoo, physical samples are collected, and tetanus inoculation at 1 year of age and booster at 2 years of age are administered. During the trapping season, some individuals have been permanently removed from the island to control for population size [16]. Annual removal strategies have varied (from no removal to up to 596 individuals removed [16], for details) and include removal events of entire social groups, as well as age-specific and sex-specific removal events. Within such structure (age and sex), individual IDs for removal are selected at random. Visual census reports on detailed individual life histories have been recorded since 1956, and continuous information on all individuals born in the population is available since 1973. Daily censuses are taken by designated staff 5 days a week ([17], for details). Records include ID, date of birth, sex, maternal genealogy, social group, and date of death or date of permanent removal from the island.

## 2.2. Major hurricanes impacting Cayo Santiago

The CSFS has experienced the direct impact of three major hurricanes (category 3–5) since the establishment of census records in 1956: Hugo (18 September 1989), Georges (21 September 1998) and Maria (20 September 2017 [18,19]; figure 1). Hugo and Georges were category 3 hurricanes when their centres were closest to Cayo Santiago (approx. 23 km and approx. 8.4 km from CSFS, respectively). Hurricane Hugo exhibited sustained winds of approximately 201 km h$^{-1}$ [20] while hurricane Georges exhibited sustained winds of 185 km h$^{-1}$ with hurricane-force winds extending 140 km from its centre [21,22]. Hurricane Maria was a category 4 hurricane when its centre was closest to Cayo Santiago (approx. 20 km). This hurricane exhibited sustained winds of 220 km h$^{-1}$ and hurricane-force winds extending 95 km from its centre [23]. The infrastructure and vegetation of Cayo Santiago was severely damaged following all three events and, following Hurricane Maria, the isthmus connecting both cays in the island of Cayo Santiago disappeared (figure 1). Food provisioning and census taking was resumed no later than 1 and 2 days after Hugo and Georges (John Berard, personal communication on 11 April 2019), respectively, and no later than 3 days after Maria. Fresh water was assumed to be uninterrupted due to naturally formed ponding from rainfall.

## 2.3. Demographic analysis

Our demographic analysis is based on the 45-year period from 1973 to 2018. During this period, a total of 4635 female rhesus monkeys were tracked. For each year, we parametrized female-only, birth-pulse models employing post-breeding censuses [24]. Given the observed annual timing of births across the history of the population [13], we defined the annual structure in our analysis from 1 June at time $t$ to 31 May at time $t + 1$ to avoid overlap of birth seasons. During this period, females were tracked and placed into categories based on developmental and reproductive stages (figure 2). In a given year, we classified females under 3 years of age in one of three age-specific developmental stages: (i) infants (I; 0-year-olds), (ii) yearlings (Y; 1-year-olds) and (iii) juveniles (J; 2-year-olds). When reaching 3 years of age, females were then classified into adults in one of three reproductive stages: (i) non-breeders (NB), (ii) failed breeders (FB) and (iii) successful breeders (B). Non-breeders were adult females who did not give birth to a female at time $t$. Failed breeders were adult females who gave birth to a female offspring at time $t$; however, the offspring did not live to be 1 year of age. Breeders were adult females who had a female offspring at time $t$ that survived to 1 year of age (recruitment). In this population, mother–infant relationships decrease with age [25], females can resume mating during the following year, and a boost in survival is observed once individuals reach the yearling stage [12]. Adult females transitioned among these three categories until death or until being right

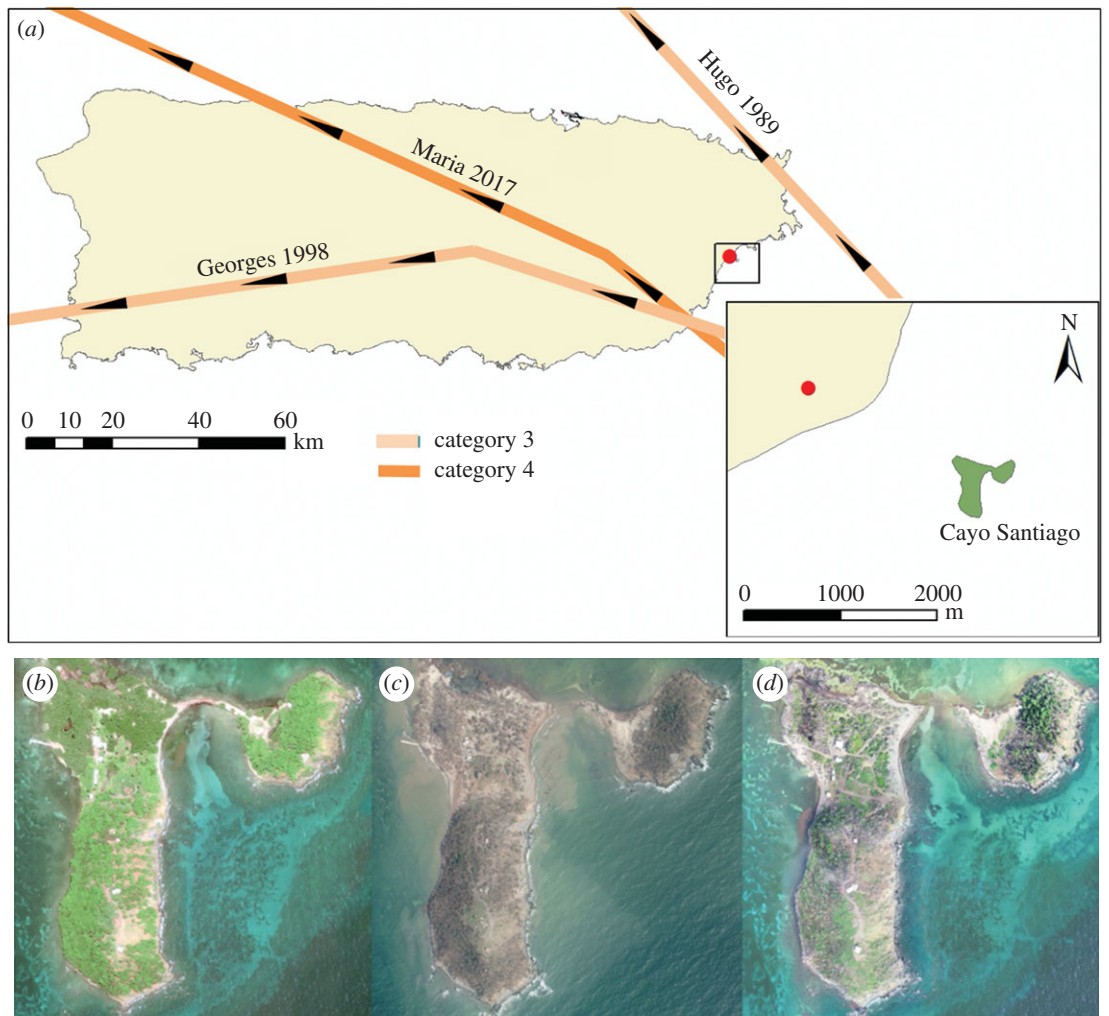

**Figure 1.** Tracks of Hurricanes Hugo, Georges and Maria (*a*); and satellite images of Cayo Santiago Field Station showing tree cover in 2017 before Hurricane Maria (*b*), greater than 60% tree defoliation (brown colour shows defoliated tree trunks) and disappearance of isthmus 90 days after Hurricane Maria (20 September 2017) (*c*), and aerial image showing partial tree cover recovery 477 days after the event (*d*). Qualitative information of hurricane tracks was extracted from the National Oceanic Atmospheric Administration (NOAA, https://coast.noaa.gov/hurricanes/). Image credit: NOAA (satellite images), Michelle Skrabut La Pierre (aerial image).

censored due to being alive or removed from the population. Although transitions from J to B or from J to FB are rare, they are expected to be non-zero as a small portion of 3-year-old females may give birth [12,26].

Using this data, we parametrized 44 matrix models, $\mathbf{Q}_t$, one for each annual period $t$, with discrete stage transitions $q_{ijt}$ (figure 2). In each annual model, individuals transition between stage $j$ at time $t$ and stage $i$ at time $t+1$ and have stage-specific survival rates defined as $\sum_{i=1}^{n} q_{ijt} = 1 - d_{jt}$, where $d_{jt}$ is the stage-specific mortality in stage $j$ and period $t$, and $n$ is the number of stages. Since only stage B individuals contributed to reproduction, their fertility was set to 1 (100%) while fertility of stages NB and FB were set to 0. Following this, survival of infants was set to 1, as only surviving infants were recruited into the population. Thus, each matrix element ($q_{i,j}$) corresponds to stage-specific vital rates; the probability of surviving and transitioning into another stage; the probability of surviving and staying in the same stage (stasis); and fertility ($f_B$, figure 2). Note that our model considers reproduction to occur before the stage transitions and thus, NB or FB present fertility = 0.

For each matrix model, we determined the mean population growth rate ($\lambda$, the population long-term fitness), the stable stage distribution ($w$) and the reproductive value vector ($v$), by computing the dominant eigenvalue and corresponding right and left eigenvectors, respectively [24]. To test for differences between individual transitions generated during non-hurricane and hurricane years, a bootstrap was performed using both the non-hurricane year dataset and the hurricane year dataset.

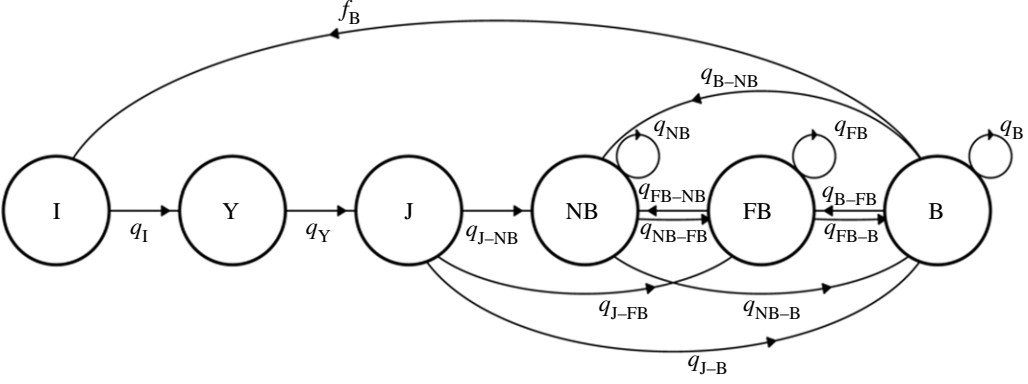

**Figure 2.** Life cycle graph of the six stage transitions based on developmental (I, infant; Y, yearling; J, juvenile) and reproductive stages (NB, non-breeder; FB, failed breeder; B, successful breeder). Each coefficients in the life cycle graph, $q_{ij}$, enters into the projection matrix **Q**.

Bootstrapping is a common resampling method to estimate measures of uncertainty for demographic parameters when using matrix models without parametric assumptions or balanced data [24]. Major hurricanes are rare extreme events, they occurred only three times in our 45-year study and thus generated unbalanced demographic datasets. However, our analysis is based on data from all females in the population and thus, our population-level information and sample sizes are appropriate for this method. For this, we combined individual stage transitions from the 41 non-hurricane years (1973–1989, 1990–1998, 1999–2017), which included a total of 18 344 individual transitions. Similarly, we combined individual stage transitions from hurricane years (1989–1990, 1998–1999, 2017–2018), which included a total of 1816 individual transitions. For each one of the datasets, individual transitions were randomly selected with replacement, until the original empirical sample size was obtained. Demographic transitions were calculated from this sample, allowing a transition matrix to be constructed and $\lambda$ estimated. This process was repeated 1000 times and 95% CI were determined from 2.5th and 97.5th percentiles of the data [24]. A two-sample Kolmogorov–Smirnov (K-S) test was performed to compare statistically both bootstrap $\lambda$ distributions.

Our demographic model incorporates stage-specific vital rates to estimate $\lambda$, a measure that synthesizes vital rates into population-level performance (i.e. population growth rate). Because $\lambda$ presents different sensitivities to each vital rate, not all environmentally induced changes in vital rates affect $\lambda$ equally. For instances, a large effect on a vital rate to which $\lambda$ is insensitive probably contributes less to variation in $\lambda$ than a smaller effect on a vital rate to which $\lambda$ is highly sensitive [27]. To determine which vital rate (stage-specific survival, stage-specific fertility) contributed the most to the observed changes in $\lambda$ during hurricane years, we conducted a life table response experiment (LTRE [27]). The LTRE is a retrospective analysis that quantifies the population-level effect of environmental factors by measuring a set of vital rates under different environmental conditions (treatments), analogous to the use of linear models to decompose the results of experiments into an additive set of treatment effects. In this way, our LTRE decomposes treatment effects (i.e. hurricane) into contributions from each of the vital rates. For this, two mean matrix models were parametrized from individual transitions; one using the non-hurricane year dataset (**G**, reference matrix) and another for the hurricane year dataset (**B**, treatment matrix) and the matrix of contributions **C** was estimated. In this way, each entry in the matrix of contributions represents a contribution of the effect of hurricanes on its corresponding matrix entry (vital rate) to the overall hurricane effect on $\lambda$.

Given density-dependent dynamics affecting fertility in Cayo Santiago females [12], we further evaluated the effects of treatment (i.e. hurricane) on mean age-specific fertility after controlling for population density (i.e. number of adult females) using generalized additive mixed models. Generalized additive mixed models are non-parametric extensions of generalized linear models that allow the evaluation of nonlinear relationships and thus, they are appropriate to model age-specific fertility in primates. For this, we used a logit function for a binary outcome of offspring birth among adults, evaluated for treatment as grouping factor and age as the smooth term. To include potential variability from maternal investment, we included offspring of both sexes in this analysis ($N = 12\,828$). To determine whether the variability in age-specific fertility was explained by major hurricanes, we

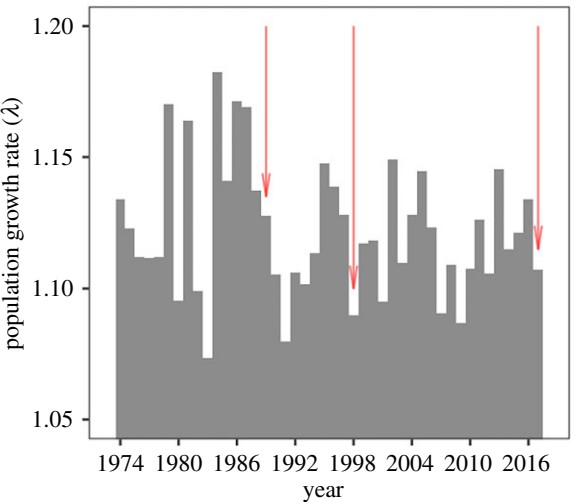

**Figure 3.** Mean annual population growth rate of Cayo Santiago rhesus macaques. During this period, three major hurricanes occurred indicated by red arrows (Hugo, 1989; Georges, 1998; Maria 2017).

fitted a series of competing models that included a linear fixed effect for each level of treatment (non-hurricane year, hurricane year) and a factor smooth interaction. To account for temporal variation in population density and because we tested the same female multiple times, we included a pair of crossed random effects; the annual number of adult females and individual ID as random intercepts. Considering all factor combinations, this resulted in a total of five competing models that were evaluated using Akaike's information criterion (AIC) for model selection [28]. All analyses were carried out using R [29], v. 3.5.1 and packages Popbio [30], and gamm4 [31]. Data and codes for the analysis are deposited in Dryad [32].

# 3. Results

## 3.1. Demographic analysis

During the 45-year period, the mean annual population growth rate ($\lambda$) was 1.122 (95% CI: 1.080, 1.171), indicating a mean annual growth of 12%, approximately, over the entire study period (figure 3). Non-hurricane years exhibited a $\lambda$ of 1.123, while hurricane years exhibited a $\lambda$ of 1.108, resulting in a 1.34% change in $\lambda$. Specifically, hurricanes Hugo (1989), Georges (1998) and Maria (2017), exhibited a $\lambda$ of 1.128, 1.090 and 1.107, respectively (red arrows in figure 3).

The demographic transitions of reproductive rhesus macaque females were altered during hurricane years as a consequence of reduced fertility. During non-hurricane years, the mean population matrix is characterized by higher rates of stasis (surviving and remaining in the same reproductive stage) among breeders and higher rates of transition toward breeders from both non-breeder and failed breeders (table 1). During hurricane years, breeder stasis and transitions toward breeders declined, while transitions to non-breeders and failed breeders were favoured. By contrast, survival and transitions among developmental stages remained the same across years.

With the exception of yearlings and failed breeders, mean mortality rate increased across stages during hurricane events (table 1 and figure 4). Juvenile mortality increased by 55%, while non-breeder and breeder mortality rates increased by 6.9% and 6.6%, respectively. By contrast, yearling mortality decreased by 3.2%, while failed breeder mortality decreased by 69.2% (figure 4; note that infant mortality is not shown as it equals 0). However, the observed variability in mortality rates also increased substantially during hurricane events, suggesting no overall difference in mortality due to hurricane events.

## 3.2. Stable stage distribution and reproductive value

The stable stage distribution (SSD; $w$) computed from $\mathbf{Q}_t$ correlated with the observed stage-specific distribution of females as reported previously ([33], electronic supplementary material), indicating

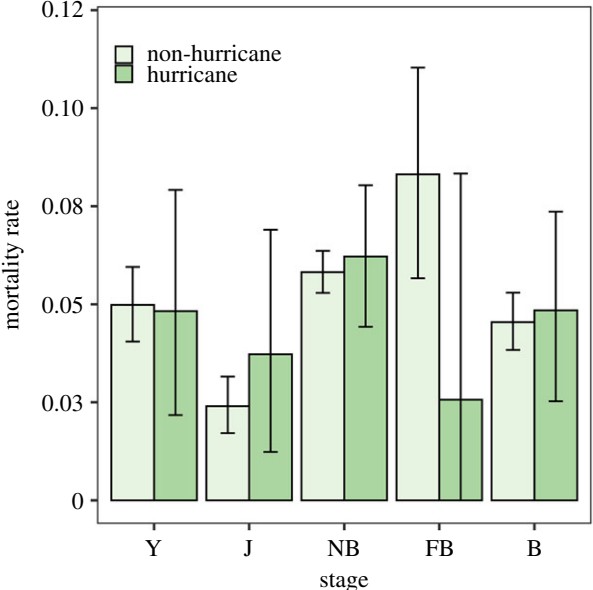

**Figure 4.** Stage-specific mortality rate of Cayo Santiago rhesus macaques for non-hurricane and hurricane years. Y, yearling; J, juvenile; NB, non-breeders; FB, failed breeders; B, successful breeders. Error bars represent 95% confidence intervals.

**Table 1.** Mean projection matrix of Cayo Santiago rhesus macaques during non-hurricane and hurricane years. Note: non-hurricane years are defined from 1974–1989, 1990–1998 and 1999–2017. Hurricane years correspond to 1989–1990, 1998–1999 and 2017–2018. Stages I, infant; Y, yearling; J, juvenile; NB, non-breeder; FB, failed breeder; B, successful breeder.

| | reproductive stages at period $t$ | | | | | |
|---|---|---|---|---|---|---|
| period at $t+1$ | I | Y | J | NB | FB | B |
| non-hurricane | | | | | | |
| I | 0 | 0 | 0 | 0 | 0 | 1.000 |
| Y | 1.000 | 0 | 0 | 0 | 0 | 0 |
| J | 0 | 0.950 | 0 | 0 | 0 | 0 |
| NB | 0 | 0 | 0.967 | 0.604 | 0.533 | 0.602 |
| FB | 0 | 0 | 0.003 | 0.040 | 0.071 | 0.036 |
| B | 0 | 0 | 0.006 | 0.298 | 0.313 | 0.317 |
| hurricane | | | | | | |
| I | 0 | 0 | 0 | 0 | 0 | 1.000 |
| Y | 1.000 | 0 | 0 | 0 | 0 | 0 |
| J | 0 | 0.952 | 0 | 0 | 0 | 0 |
| NB | 0 | 0 | 0.963 | 0.606 | 0.667 | 0.633 |
| FB | 0 | 0 | 0 | 0.054 | 0.051 | 0.045 |
| B | 0 | 0 | 0 | 0.278 | 0.256 | 0.273 |

that our annual models based on stable stage theories accurately represent the observed changes in stage distributions. The mean stable stage distribution $w$ for non-hurricane years (**G**) was biased towards NB females with 14.8% I, 13.1% Y, 11.1% J, 42.3% NB, 2.19% FB and 16.6% B (figure 5*a*). During hurricane years (**B**), $w$ showed a decrease in developmental stages and an increase in NB and FB; 14.0% I, 12.6% Y, 10.8% J, 44.1% NB, 2.89% FB, and 15.5% B. During non-hurricane years, $v$ was 1.00, 1.12, 1.34, 1.54, 1.52 and 2.46 for I, Y, J, NB, FB and B, respectively. During hurricane years $v$ showed a decrease in

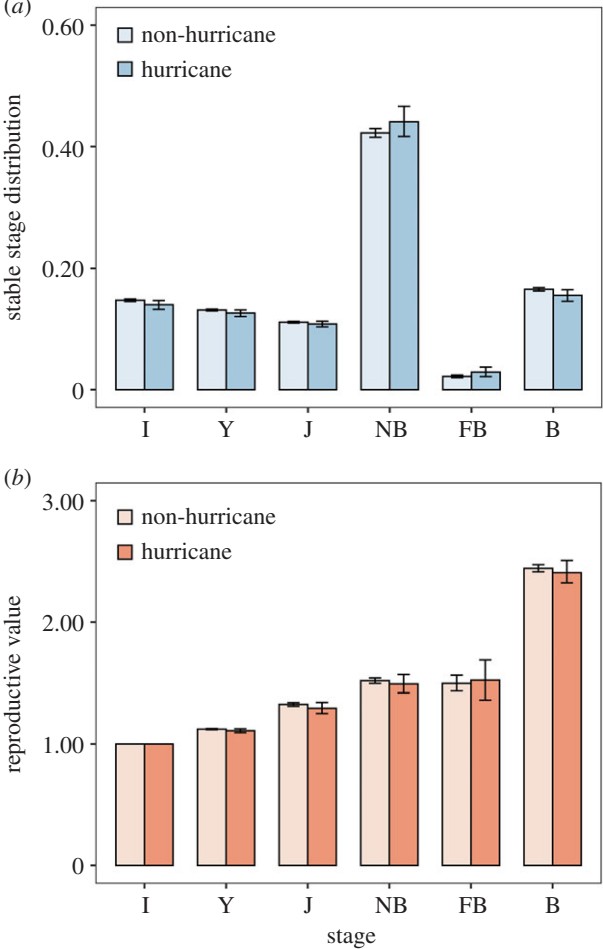

**Figure 5.** Stable stage distribution (*a*) and reproductive value (*b*) of Cayo Santiago rhesus macaques for non-hurricane and hurricane years. I, infants; Y, yearling; J, juvenile; NB, non-breeders; FB, failed breeders; B, successful breeders. Error bars represent 95% confidence intervals.

most stages with values of 1.00, 1.11, 1.33, 1.52, 1.53 and 2.44 for I, Y, J, NB, FB and B, respectively (figure 5*b*).

## 3.3. Population-level effects of variation in vital rates during hurricane years

The frequency of $\lambda$ generated through bootstrapping the transition data from non-hurricane and hurricane years resulted in two different distributions (K-S test: $D = 0.814$, *p*-value < 0.001, $N = 2000$; figure 6*a*). During non-hurricane years, the distribution of $\lambda$ showed a mean of 1.122 (95% CI: 1.118, 1.127); while during hurricane years the distribution of $\lambda$ showed a mean of 1.110 (95% CI: 1.094, 1.125). The life table response experiment shows that this decline in $\lambda$ was mostly due to a decrease in fertility given the decrease in transitions from NB to B, followed by the decrease in B stasis (figure 7; electronic supplementary material). In fact, nearly 45% of the effect of hurricanes on $\lambda$ comes from changes in NB transitions (i.e. sum of the NB column divided by the sum of the entries in the matrix of contributions **C** using absolute values), while 38% comes from B transitions. The matrix of contributions **C** shows positive and negative contributions that partially cancel each other out indicating hurricanes had a small effect on $\lambda$. Yet, the overall effect of hurricanes over $\lambda$ is negative (figure 7). Negative values in the LTRE indicate transitions that contributed to the decline in population growth rate and positive values indicate transitions that enforced population growth during hurricane years. In agreement with this, we found support for treatment-specific differences in fertility across age (cumulative model weight 0.70; figure 6*b*; see electronic supplementary material for the complete list of model weights). Non-hurricane years exhibited a higher mean fertility across ages ($\beta_{(intercept)} = 0.802$, $\beta_{(treatment)} = 0.368$, adjusted $R^2 = 0.16$; electronic supplementary material).

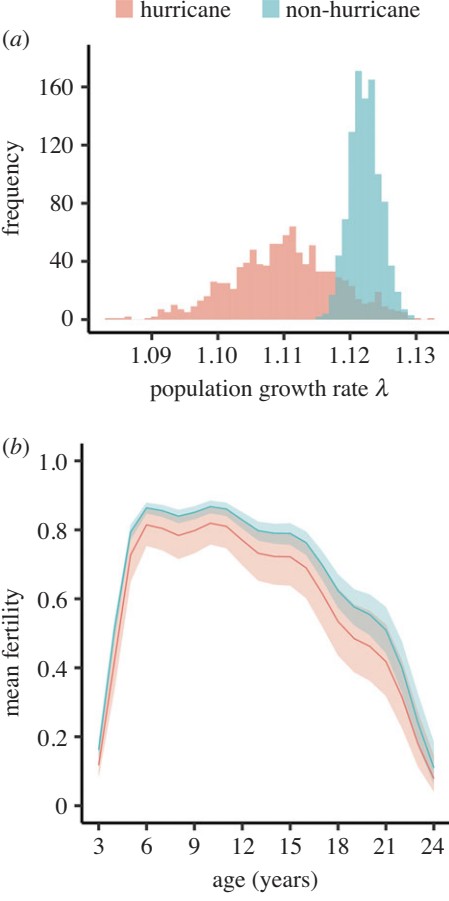

**Figure 6.** Frequency distribution of bootstrap population growth rates (*a*) and mean age-specific fertility (*b*) of Cayo Santiago rhesus macaques during non-hurricane (blue) and hurricane years (red, data from reproductive ages 3–24 shown).

## 4. Discussion

Our analysis of the Cayo Santiago rhesus macaque population reveals demographic buffering during hurricane years. Despite the observed decrease in mean population growth rate associated with a decreased fertility, the population continued to grow on average 11% annually following the impact of a hurricane. Our study shows that declines in fertility can be offset, as long as survival does not substantially decrease, such that acute extreme disturbance events do not necessarily result in detrimental effects at the population level.

The effect of major hurricanes on the population growth rate of Cayo Santiago rhesus macaque females comes mostly through decreased fertility. Our results show that hurricanes bring a negative contribution to $\lambda$ from the decreased advantage in adult reproductive output during non-hurricane years. Specifically, the decreased proportion of non-breeders transitioning to breeders, followed closely by the decrease in remaining breeders in consecutive years (B stasis), contributed the most to the decrease in $\lambda$. Changes in Cayo Santiago female fertility may involve density-dependent mechanisms [12,16]. After controlling for density effects, our analysis shows that major hurricanes also contribute to the population dynamics of Cayo Santiago monkeys through reductions in mean fertility across the lifespan. The fact that major hurricanes are rare, occurring only three times during our 45-year study period, and yet we were able to identify treatment effects, highlights the major demographic consequences these climatic events can produce. The mechanisms underlying hurricane-induced decreased fertility may involve deficient energy budgets and high levels of social stress following significant deforestation and subsequent food shortage [5,34,35]. For instance, black-and-white lemurs in Manombo forest suffered from a lack of successful reproduction during the three years following a major hurricane potentially due to food limitation [34]. The latter is expected from the link between energy acquisition and storage and the ability of females to conceive or sustain pregnancies [36,37], a common pattern across mammals. Similarly, the energy demand from chronic physiological stress and

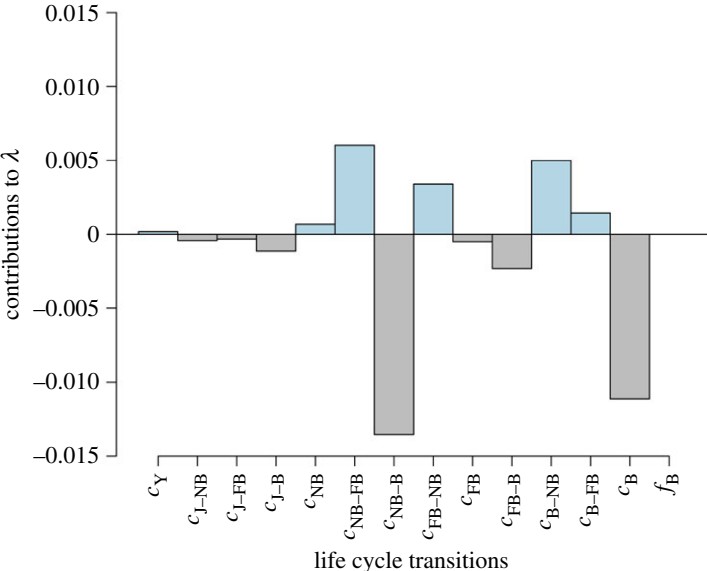

**Figure 7.** Contributions of vital rates of Cayo Santiago rhesus macaque females to changes in population growth rate during hurricane years. Values are extracted from the matrix of contribution **C** of the life table response experiment (LTRE; electronic supplementary material).

persistently high cortisol levels associated with significant habitat change can result in decreased reproductive rates [5,35]. Our study population is provisioned with (ad libitum) food, yet individuals are known to spend 50% of feeding time on natural vegetation [14]. Specially, our study population presents social stratification, which is related to differential access to food, and thus the amount consumed by individuals varies significantly according to their social rank [15]. Moreover, Cayo Santiago suffered a 60–90% canopy loss following each hurricane event (Matthew Kessler 2019, personal communication) [38]. Such significant change in the landscape probably caused substantial diet and daily-life behavioural changes that lasted long enough (i.e. months to years, figure 1) to affect demographic fates of individuals after each event, contributing to the observed decreased fertility.

Despite the decrease in mean fertility among females, the Cayo Santiago population kept growing at a relatively high rate (11%) annually exhibiting only a 1.34% decline in $\lambda$ after major hurricanes. When compared to studies of wild rhesus populations, Cayo Santiago exhibits higher than average vital rates [39–41]. Previous studies in Cayo Santiago females also show low temporal variability in survival, including hurricane years 1989 and 1998 [12]. Such dynamics are reflected in the small changes observed in the stable stage distribution during hurricane years. Consequently, the reproductive value vector decreased only slightly for breeders and non-breeders during hurricane years. Yet, the Cayo Santiago population follows demographic patterns observed in the wild such as higher sensitivity in adult survival. Long-lived mammal populations often exhibit a higher contribution of survival to population growth rate (high sensitivity), relative to that of fertility [12,42], avoiding jeopardizing their survival at the cost of reproduction [43]. This is a common finding; population growth rate is most sensitive to adult survival relative to other demographic traits such as survival of developmental stages or age at reproduction [44,45]. Thus, any environmental event affecting fertility temporarily is expected to have less effect on $\lambda$ than an event affecting survival due to multiple reproductive seasons across the lifespan, and thus, high potential for recovery. Furthermore, animals with slow life histories and longer generation times linked to well-developed brains may respond to environmental change through behavioural flexibility [45,46], including shifting diets following dramatic changes in the landscape [47–50]. Many primates, specially members of island or coastal populations, are subjected to catastrophic large-scale habitat modification due to tropical storms or hurricanes, making behavioural flexibility a common aspect of their selective history [51].

Hurricanes are also linked to up to 50% declines in primate populations through increased mortality [8,52,53]. While the hurricane-force winds may play a direct role in the reported high mortality rates through injuries, other factors related to the aftermath of the hurricane, such as altered habitats and up to 100% canopy loss, are thought to have an additional contribution to population decline [54,55]. However, our analysis did not find major contributions to the population growth rate from reduced survival. Most likely, the prompt resumption of food provisioning allowed a relatively high portion of the population to allocate energy to survival and continued increase in growth.

Finally, hurricanes can affect primate demography through extended social network disorganization. For instance, the multiple benefits of group living, including higher foraging efficiency and improved resource defence can be interrupted by extreme climatic events [47,48]. Following a hurricane event, primate populations may exhibit social group disruption and an increase in the number of solitary individuals and smaller social groups with no stability—steady group composition and range—up to three months following a hurricane event [8,9,55]. However, following Hurricane Maria, a higher density in proximity networks and less isolation of individuals was reported in Cayo Santiago [56,57]. Strong social ties with other members of the population may act to lower cortisol levels or heighten immune responses following stressful events [58–60].

Our analysis quantifies the demographic mechanisms shaping population fitness in a long-lived non-human primate population following extreme climatic events and demonstrates such mechanism is mediated through the maintenance of survival. The study reflects previous findings in other long-lived animal populations in which life-history traits with the highest contribution to population fitness ($\lambda$) exhibit low variability (i.e. survival for Cayo Santiago monkeys), maintaining population growth [61]. Our analysis also provides new information on resistance or potential adaptive mechanisms following extreme events. The Cayo Santiago population apparent strategy to maintain growth during hurricane years involves suppressing mean fertility. If survival is maintained at the expense of such suppression, our study can serve as a route for new questions regarding the role of differential life-history strategies and trade-offs in the adaptive potential of primate populations. Finally, our study supports evidence claiming the need to incorporate detailed life-history traits—stage-specific vital rates—on population studies in order to make accurate assessments of population dynamics and viability following extreme disturbance events [6].

# 5. Future directions

Major hurricanes are rare climatic events characterized by a set of acute environmental factors acting over short periods of time. Their extreme nature makes demographic model parametrization and power challenging. As data from long-term studies accumulates, new opportunities for alternative ways of modelling these events become possible. Future directions in modelling the effects of hurricanes on demography include modelling vital rates as a function of hurricane-associated variables (covariates; e.g. canopy cover, windspeed, rainfall) ([62], for a similar approach in primates). However, these analyses can be very sensitive to assumptions and, given the rare frequency of hurricanes, data may remain scarce. Defining the environment as a stochastic component and evaluating the variance in vital rate parameters during hurricanes and non-hurricane years ([45], for a similar approach in primates) are other potential venues to evaluate hurricanes as drivers of population performance. Measures of individual stochasticity [63] and dynamic heterogeneity [64] will allow us to evaluate the role of hurricanes as drivers of individual life histories. Lastly, extending these approaches to two-sex models could provide additional information on the demographic effects of major hurricanes on non-human primate populations.

Data accessibility. Demographic data used in this study are available in the Dryad Digital Repository: https://dx.doi.org/10.5061/dryad.5qfttdz2b [32].

Authors' contributions. R.H.-P. designed, coordinated and supervised the research. D.O.M. performed all data analyses and created all figures. A.V.R.-L. provided support on data sharing and access, and interpretation of Cayo Santiago archives. R.H.-P., D.O.M., A.V.R.-L., U.K.S. and K.L.G. commented in the study design and interpretation, and contributed in writing the manuscript.

Competing interests. The authors declare no competing interests.

Funding. D.O.M. was supported by the University of Richmond, Arts and Science Summer Fellowship. R.H.-P. was partly supported by the University of Richmond School of Arts & Sciences Dean's Office. R.H.-P. was partly supported by the University of Richmond, Dean's Office.

Acknowledgements. We thank members of Cayo Santiago and the Caribbean Primate Research Center who contributed to census data collection. The content of the publication is the sole responsibility of the authors and does not necessarily represent the official views of the National Center for Research Resources, ORIP, or UPR. Cayo Santiago is supported by the Office of Research Infrastructure Programs (ORIP) of the National Institute of Health, grant 2 P40 OD012217, and the University of Puerto Rico (UPR) Medical Sciences Campus. We also thank Matt J. Kessler and John Berard for their personal communications on the post-hurricane management of the colony following hurricanes Hugo and Georges, as well as Diana L. Delgado for providing the map of the study site, and Michelle Lapierre Skrabut for the Cayo Santiago aerial images.

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
