## [Reviewer comments · Royal Society Open Science]

Review History

RSOS-200173.R0 (Original submission)

Review form: Reviewer 1

Is the manuscript scientifically sound in its present form?

Yes

Are the interpretations and conclusions justified by the results?

Yes

Is the language acceptable?

Yes

Do you have any ethical concerns with this paper?

No

Have you any concerns about statistical analyses in this paper?

No

Recommendation?

Accept with minor revision (please list in comments)

Comments to the Author(s)

This is a solid analysis and a well-written manuscript so I don't have many critical comments. It is generally acceptable for publication as is, though I do suggest (not mandate) some additions/changes that would make this manuscript perhaps better. I also have a few questions about the analysis, but these questions are simply to make sure you have robustly parameterized your matrix population model (MPM).

--It might be nice to include a life cycle graph as a figure. It took me awhile to figure out the life cycle transitions and once I did, this led me to a question/comment: The life cycle (as reflected in the projection matrix in Table 1) appears to show that a juvenile must transition to an adult female non-breeder before transitioning to failed breeder or breeder. That is, one can only go from stage 3 to stage 4, but not 3 5 or 36. Is this a biologically defensible representation of macaque fertility? Is it possible for a juvenile to transition to a failed breeder or breeder across a projection interval (year)? If females can only go from stage 3 to 4, but not stage 3 to 5 or 3 to 6, that's fine...just wanted you to double-check this.

--Another comment regarding life cycle...I expected to see three entries in the top-row of the projection matrix (the F terms in life cycle graph terminology, following Caswell 2001) corresponding to fertility (F4, F5, F6). Instead there is only 1 entry with a value of 1 at 1,6 of the projection matrix. The life cycle graph shows that one can make the following transitions 46, 56, and 66, which means that, as far as fertilities go, one would expect an entry in the 1,4 position of the projection matrix corresponding to a non-breeder who doesn't have an offspring at time t, but has an offspring at t+1. Same goes for failed breeders, the 56 transition reflects females who failed in year t, but have an offspring in year t+1, thus requiring an entry in the 1,5 position of the projection matrix. Same for the 66 transition and corresponding to the F6 entry. For example, in Brault/Caswell's 1993 paper on killer whales (in Ecology), there is a F2 entry corresponding to a juvenile who doesn't have a calf at time t, but does have one at time t+1 as it makes the transition to maternity, and there is a F3 entry corresponding to mothers who repeatedly reproduce from year to year. In any case, I just wanted to bring this to your attention--I could be wrong--so feel free to ignore if you feel you have parameterized fertility correctly.

--In the discussion section, you could bolster some of your points by bringing in more comparative data from other primate populations. For example, in lines 294-296, you write that population growth rate is generally influenced more by survival than by fertility--this is a common finding across primates (my own review of this was in 2011 in Yearbook of Physical Anthropology) and other mammals that doesn't just pertain to climatic events: population growth rate is most sensitive to adult survival in many demographic analyses. And the difference in vital rates, stable stage distribution, reproductive value, and lambda is quite minimal between hurricane versus non-hurricane years, so bringing some comparative data along these lines might be useful to point out. More generally, the population growth rate of the Cayo population is quite high even during hurricane years, at about +10% per year and this is much higher than estimates for wild primates--perhaps some discussion of this is warranted(?).

--In line 297, you mention "transient dynamics" as having a little effect. I would be careful about using the phrase "transient dynamics", as you didn't formally analyze transient dynamics, so it is not clear how you determined they have a small effect.

--the LTRE could be both better described in the Methods and better analyzed/discussed in the Discussion, especially for folks (e.g., psychologists, behavioral ecologists) who are not familiar with this type of analysis.

--In the Discussion, you might briefly mention an alternative way to incorporate environmental effects in terms of how they influence the vital rates. The present manuscript builds time-dependent MPMs for Hurricane versus non-hurricane years and then compares the (average) differences in growth rate and other vital rates across the two categories. This is fine and good,

but it is also possible to model the environment directly by writing a hurricane-associated variable (rainfall, windspeed, etc.) as a covariate; in this regard, each vital rate would be a linear function of the covariate; this method is used widely (and implemented in program MARK) and also discussed by Fujiwara and Caswell in the journal *Ecology* (2002). Lawler et al., (2009 *Oecologia*) use this approach to model the effects of rainfall on vital rates in a primate population. The "covariate" approach would better allow you to actually model what you are trying to model: how hurricanes influence the vital rates. Using a MPM from a hurricane year likely subsumes all sorts of factors, both from the hurricane itself and before (e.g., feeding competition, social cohesion, dominance, etc.), and thus doesn't explicitly capture how a climatic variable directly influences a given vital rate. I'm not saying you need to do this analysis, but you could at least mention it.

--In the Discussion section, the last two sentences of the paper could be more clear. It is not clear how this analysis will provide information on "resistance or potential adaptive mechanisms opening new questions regarding the role of trade-offs between survival and reproduction...". I apologize but I'm not sure what this means? Trade-offs between survival and reproduction (in a life history sense) are set by long-term evolutionary forces and might also have clade-specific effects. (As an aside, an analysis of the trade-off between survival and reproduction in Cayo Santiago macaques was conducted by Greg Blomquist (*Biology Letters*, 2009), who showed that there was no phenotypic trade-off, but there was an additive genetic correlation that reflected a trade-off between longevity and reproduction). I would reword this sentence and unpack it some more to enhance the clarity. The very last sentence might be better worded as well, or at the very least one could provide references for the claim that "Our study supports evidence claiming the need" to incorporate life history traits in population studies following climatic events.

In any case, as you can see, most of my comments are basically suggestions and/or queries about the analysis. This is a nice analysis. I hope you find my suggestions helpful.

Review form: Reviewer 2

Is the manuscript scientifically sound in its present form?

No

Are the interpretations and conclusions justified by the results?

No

Is the language acceptable?

Yes

Do you have any ethical concerns with this paper?

No

Have you any concerns about statistical analyses in this paper?

Yes

Recommendation?

Major revision is needed (please make suggestions in comments)

Comments to the Author(s)

Review of 'Hurricane-induced demographic changes in a nonhuman primate population'
I enjoyed reading this well-written manuscript about the dynamics of an introduced rhesus macaque populations on a 15ha island of the coast of Puerto Rico. The authors ask how 3 years in which the island was hit by major hurricanes differed from the other 41 years in this impressively

longitudinal demographic study. However, I also have major questions about the population models, analyses and interpretation.

First of all, I see little to no evidence that the hurricanes have affected the survival, breeding or population size of the macaques. The authors claim that there is a significant difference in projected population growth rate between non-hurricane and hurricane years ($\lambda = 1.123$ vs 1.108). But given that there were only 3 hurricane years I am not convinced that this is a meaningful difference in λ . Figures 2 and 5 show that there is strong overlap in the ranges of λ values (to the degree we can talk about a range based on 3 years). The authors apply a nonparametric KS test based on the bootstrapped λ values, but such an approach is highly problematic. P values directly depend on the number of bootstraps. Given high enough number of bootstraps, any small difference in mean λ can be 'proven' significant. In addition, the bootstrapping is not stratified within years, meaning that interannual variation in environmental conditions, population size, sample size and demographic rates is ignored when comparing dynamics in non-hurricane and hurricane years.

While we are not shown how population size fluctuates over the study period, the authors state in the Discussion that population sizes were not much different between hurricane and non-hurricane years, which could be interpreted as a sign of strong density-dependence, external population regulation and/or absence of effect of hurricanes. Without further analyses it is hard to distinguish between these factors. The authors do state that survival rates are unaffected by hurricane years (here statistical tests would be appropriate but missing), but that there are effects on breeding probabilities. Again, these differences are not tested for statistical significance. For that purpose, and also to enable easier biological interpretation, it would be better if the authors present underlying vital rates (e.g. breeding probability conditional on survival) rather than only matrix element values. Given that only 3 hurricane years were observed and the considerable variation in λ , I hardly expect significant differences in breeding probabilities among years caused by hurricanes.

Are there additional ways in which the authors can unveil the alleged demographic effects of hurricanes? The authors claim that hurricanes do not directly cause additional mortality, but that the most likely longer-term effect acts through defoliation. To test their hypothesis, I urge the authors to analyse annual variation in vital rates and λ as a function of annual estimates of canopy cover. Using a continuous explanatory variable would also allow for understanding variation among years better.

Without clear patterns to show, readers are left to wonder whether or not the potential mechanisms of population response to hurricanes are relevant or not, and more importantly whether this study system is suitable for answering those questions. The population is fed regularly by humans and population sizes are regulated as well.

The constructed population models project a mean annual growth of 12%. That would mean that when starting with the original 409 monkeys, 44 years later one would count nearly 60 thousand animals. Clearly these models do not take population regulation (tetanus inoculation and removal by humans) into account. But readers have no way of assessing whether 12% growth is realistic, how much individuals are removed each year, nor how strong density dependence is in this population. This makes it hard to be confident that the constructed population models are good representations of the population dynamics, which is important as the authors attempt to study relatively subtle effects. As an additional test, do the models project realistic life spans?

64-65 please explain this hypothesized trade-off in more detail

77-79 please be more precise in formulating LTRE analyses

102 why were the macaques introduced in 1938?

109 why 'commercial'?

115 how were animals caught? Age-specific?

119 # removed annually? Target numbers?

122 unclear how individuals are censused daily

152 I understand sons are not counted for population growth, but having sons would affect maternal investment compared to non-breeders, right?

162 I do not understand this part well. So adults are classified as breeders (at time t) when they are going to have a live daughter next year ($t+1$) given that is also alive at time $t+1$? Table 1 does suggest that. But is this interpretation correct? And what does definition of 'breeder' mean for the analyses of breeding probabilities and the tested trade-offs between survival and reproduction? Reproductive investment are done also when offspring do not make it to year $t+2$. Is that fully captured by the failed breeder class? My confusion also stems from a definition of breeder that apparently relies on events spread out over 2 years, while the population model has a time step of 1 year.

229-230 how much buffering is there compared to the net effect?

613 I highly appreciate that R code with the constructed matrices are provided.

Decision letter (RSOS-200173.R0)

03-Apr-2020

Dear Dr Hernández-Pacheco,

The editors assigned to your paper ("Hurricane-induced demographic changes in a nonhuman primate population") have now received comments from reviewers. We would like you to revise your paper in accordance with the referee and Associate Editor suggestions which can be found below (not including confidential reports to the Editor). Please note this decision does not guarantee eventual acceptance.

Please submit a copy of your revised paper before 26-Apr-2020. Please note that the revision deadline will expire at 00.00am on this date. If we do not hear from you within this time then it will be assumed that the paper has been withdrawn. In exceptional circumstances, extensions may be possible if agreed with the Editorial Office in advance. We do not allow multiple rounds of revision so we urge you to make every effort to fully address all of the comments at this stage. If deemed necessary by the Editors, your manuscript will be sent back to one or more of the original reviewers for assessment. If the original reviewers are not available, we may invite new reviewers.

- Data accessibility

If you wish to submit your supporting data or code to Dryad (<http://datadryad.org/>), or modify your current submission to dryad, please use the following link:
<http://datadryad.org/submit?journalID=RSOS&manu=RSOS-200173>

- Competing interests

- Authors' contributions

- Acknowledgements

- Funding statement

on behalf of Dr Ottar Bjørnstad (Associate Editor) and Pete Smith (Subject Editor)
 openscience@royalsociety.org

Associate Editor's comments (Dr Ottar Bjørnstad):

Associate Editor: 1

Comments to the Author:

Thank you for submitting your manuscript to Royal Society Open Science. Following peer review, we have received two referee reports on your manuscript. The referees raised several concerns with regards to your manuscript, particularly the Discussion section. Upon resubmission, please ensure you respond to all comments raised by the referees accordingly.

Comments to Author:

Reviewers' Comments to Author:

Reviewer: 1

Comments to the Author(s)

This is a solid analysis and a well-written manuscript so I don't have many critical comments. It is generally acceptable for publication as is, though I do suggest (not mandate) some additions/changes that would make this manuscript perhaps better. I also have a few questions about the analysis, but these questions are simply to make sure you have robustly parameterized your matrix population model (MPM).

--It might be nice to include a life cycle graph as a figure. It took me awhile to figure out the life cycle transitions and once I did, this led me to a question/comment: The life cycle (as reflected in the projection matrix in Table 1) appears to show that a juvenile must transition to an adult female non-breeder before transitioning to failed breeder or breeder. That is, one can only go from stage 3 to stage 4, but not 3 5 or 36. Is this a biologically defensible representation of macaque fertility? Is it possible for a juvenile to transition to a failed breeder or breeder across a projection interval (year)? If females can only go from stage 3 to 4, but not stage 3 to 5 or 3 to 6, that's fine...just wanted you to double-check this.

--Another comment regarding life cycle...I expected to see three entries in the top-row of the projection matrix (the F terms in life cycle graph terminology, following Caswell 2001) corresponding to fertility (F4, F5, F6). Instead there is only 1 entry with a value of 1 at 1,6 of the projection matrix. The life cycle graph shows that one can make the following transitions 46, 56, and 66, which means that, as far as fertilities go, one would expect an entry in the 1,4 position of the projection matrix corresponding to a non-breeder who doesn't have an offspring at time t, but has an offspring at t+1. Same goes for failed breeders, the 56 transition reflects females who failed in year t, but have an offspring in year t+1, thus requiring an entry in the 1,5 position of the projection matrix. Same for the 66 transition and corresponding to the F6 entry. For example, in Brault/Caswell's 1993 paper on killer whales (in Ecology), there is a F2 entry corresponding to a juvenile who doesn't have a calf at time t, but does have one at time t+1 as it makes the transition to maternity, and there is a F3 entry corresponding to mothers who

repeatedly reproduce from year to year. In any case, I just wanted to bring this to your attention--I could be wrong--so feel free to ignore if you feel you have parameterized fertility correctly.

--In the discussion section, you could bolster some of your points by bringing in more comparative data from other primate populations. For example, in lines 294-296, you write that population growth rate is generally influenced more by survival than by fertility--this is a common finding across primates (my own review of this was in 2011 in *Yearbook of Physical Anthropology*) and other mammals that doesn't just pertain to climatic events: population growth rate is most sensitive to adult survival in many demographic analyses. And the difference in vital rates, stable stage distribution, reproductive value, and lambda is quite minimal between hurricane versus non-hurricane years, so bringing some comparative data along these lines might be useful to point out. More generally, the population growth rate of the Cayo population is quite high even during hurricane years, at about +10% per year and this is much higher than estimates for wild primates--perhaps some discussion of this is warranted(?).

--In line 297, you mention "transient dynamics" as having a little effect. I would be careful about using the phrase "transient dynamics", as you didn't formally analyze transient dynamics, so it is not clear how you determined they have a small effect.

--the LTRE could be both better described in the Methods and better analyzed/discussed in the Discussion, especially for folks (e.g., psychologists, behavioral ecologists) who are not familiar with this type of analysis.

--In the Discussion, you might briefly mention an alternative way to incorporate environmental effects in terms of how they influence the vital rates. The present manuscript builds time-dependent MPMs for Hurricane versus non-hurricane years and then compares the (average) differences in growth rate and other vital rates across the two categories. This is fine and good, but it is also possible to model the environment directly by writing a hurricane-associated variable (rainfall, windspeed, etc.) as a covariate; in this regard, each vital rate would be a linear function of the covariate; this method is used widely (and implemented in program MARK) and also discussed by Fujiwara and Caswell in the journal *Ecology* (2002). Lawler et al., (2009 *Oecologia*) use this approach to model the effects of rainfall on vital rates in a primate population. The "covariate" approach would better allow you to actually model what you are trying to model: how hurricanes influence the vital rates. Using a MPM from a hurricane year likely subsumes all sorts of factors, both from the hurricane itself and before (e.g., feeding competition, social cohesion, dominance, etc.), and thus doesn't explicitly capture how a climatic variable directly influences a given vital rate. I'm not saying you need to do this analysis, but you could at least mention it.

--In the Discussion section, the last two sentences of the paper could be more clear. It is not clear how this analysis will provide information on "resistance or potential adaptive mechanisms opening new questions regarding the role of trade-offs between survival and reproduction...". I apologize but I'm not sure what this means? Trade-offs between survival and reproduction (in a life history sense) are set by long-term evolutionary forces and might also have clade-specific effects. (As an aside, an analysis of the trade-off between survival and reproduction in Cayo Santiago macaques was conducted by Greg Blomquist (*Biology Letters*, 2009), who showed that there was no phenotypic trade-off, but there was an additive genetic correlation that reflected a trade-off between longevity and reproduction). I would reword this sentence and unpack it some more to enhance the clarity. The very last sentence might be better worded as well, or at the very least one could provide references for the claim that "Our study supports evidence claiming the need" to incorporate life history traits in population studies following climatic events.

In any case, as you can see, most of my comments are basically suggestions and/or queries about the analysis. This is a nice analysis. I hope you find my suggestions helpful.

Reviewer: 2

Comments to the Author(s)

Review of 'Hurricane-induced demographic changes in a nonhuman primate population'

I enjoyed reading this well-written manuscript about the dynamics of an introduced rhesus macaque populations on a 15ha island of the coast of Puerto Rico. The authors ask how 3 years in which the island was hit by major hurricanes differed from the other 41 years in this impressively longitudinal demographic study. However, I also have major questions about the population models, analyses and interpretation.

First of all, I see little to no evidence that the hurricanes have affected the survival, breeding or population size of the macaques. The authors claim that there is a significant difference in projected population growth rate between non-hurricane and hurricane years ($\lambda = 1.123$ vs 1.108). But given that there were only 3 hurricane years I am not convinced that this is a meaningful difference in λ . Figures 2 and 5 show that there is strong overlap in the ranges of λ values (to the degree we can talk about a range based on 3 years). The authors apply a nonparametric KS test based on the bootstrapped λ values, but such an approach is highly problematic. P values directly depend on the number of bootstraps. Given high enough number of bootstraps, any small difference in mean λ can be 'proven' significant. In addition, the bootstrapping is not stratified within years, meaning that interannual variation in environmental conditions, population size, sample size and demographic rates is ignored when comparing dynamics in non-hurricane and hurricane years.

While we are not shown how population size fluctuates over the study period, the authors state in the Discussion that population sizes were not much different between hurricane and non-hurricane years, which could be interpreted as a sign of strong density-dependence, external population regulation and/or absence of effect of hurricanes. Without further analyses it is hard to distinguish between these factors. The authors do state that survival rates are unaffected by hurricane years (here statistical tests would be appropriate but missing), but that there are effects on breeding probabilities. Again, these differences are not tested for statistical significance. For that purpose, and also to enable easier biological interpretation, it would be better if the authors present underlying vital rates (e.g. breeding probability conditional on survival) rather than only matrix element values. Given that only 3 hurricane years were observed and the considerable variation in λ , I hardly expect significant differences in breeding probabilities among years caused by hurricanes.

Are there additional ways in which the authors can unveil the alleged demographic effects of hurricanes? The authors claim that hurricanes do not directly cause additional mortality, but that the most likely longer-term effect acts through defoliation. To test their hypothesis, I urge the authors to analyse annual variation in vital rates and λ as a function of annual estimates of canopy cover. Using a continuous explanatory variable would also allow for understanding variation among years better.

Without clear patterns to show, readers are left to wonder whether or not the potential mechanisms of population response to hurricanes are relevant or not, and more importantly whether this study system is suitable for answering those questions. The population is fed regularly by humans and population sizes are regulated as well.

The constructed population models project a mean annual growth of 12%. That would mean that when starting with the original 409 monkeys, 44 years later one would count nearly 60 thousand animals. Clearly these models do not take population regulation (tetanus inoculation and removal by humans) into account. But readers have no way of assessing whether 12% growth is realistic, how much individuals are removed each year, nor how strong density dependence is in this population. This makes it hard to be confident that the constructed population models are good representations of the population dynamics, which is important as the authors attempt to study relatively subtle effects. As an additional test, do the models project realistic life spans?

64-65 please explain this hypothesized trade-off in more detail

77-79 please be more precise in formulating LTRE analyses

102 why were the macaques introduced in 1938?

109 why 'commercial'?

115 how were animals caught? Age-specific?

119 # removed annually? Target numbers?

122 unclear how individuals are censused daily

152 I understand sons are not counted for population growth, but having sons would affect maternal investment compared to non-breeders, right?

162 I do not understand this part well. So adults are classified as breeders (at time t) when they are going to have a live daughter next year ($t+1$) given that is also alive at time $t+1$? Table 1 does suggest that. But is this interpretation correct? And what does definition of 'breeder' mean for the analyses of breeding probabilities and the tested trade-offs between survival and reproduction? Reproductive investment are done also when offspring do not make it to year $t+2$. Is that fully captured by the failed breeder class? My confusion also stems from a definition of breeder that apparently relies on events spread out over 2 years, while the population model has a time step of 1 year.

229-230 how much buffering is there compared to the net effect?

613 I highly appreciate that R code with the constructed matrices are provided.

Author's Response to Decision Letter for (RSOS-200173.R0)

See Appendix A.

RSOS-200173.R1 (Revision)

Review form: Reviewer 1

Is the manuscript scientifically sound in its present form?

Yes

Are the interpretations and conclusions justified by the results?

Yes

Is the language acceptable?

Yes

Do you have any ethical concerns with this paper?

Yes

Have you any concerns about statistical analyses in this paper?

No

Recommendation?

Accept as is

Comments to the Author(s)

This is a very impressive and thorough revision that has addressed the comments by me and another reviewer. I think this revision is publishable.

Decision letter (RSOS-200173.R1)

Dear Dr Hernández-Pacheco,

It is a pleasure to accept your manuscript entitled "Hurricane-induced demographic changes in a nonhuman primate population" in its current form for publication in Royal Society Open Science. The comments of the reviewer(s) who reviewed your manuscript are included at the foot of this letter.

Please ensure that you send to the editorial office an editable version of your accepted manuscript, and individual files for each figure and table included in your manuscript. You can send these in a zip folder if more convenient. Failure to provide these files may delay the processing of your proof.

Kind regards,

Andrew Dunn

on behalf of Prof Pete Smith (Subject Editor)

Reviewer comments to Author:

Reviewer: 1

Comments to the Author(s)

This is a very impressive and thorough revision that has addressed the comments by me and another reviewer. I think this revision is publishable.

Appendix A

Jeremy Sanders
Editor-in-chief
Royal Society Open Science

Thank you very much for the positive evaluation and for the invitation to submit a revised version of the attached manuscript entitled “Hurricane-induced demographic changes in a nonhuman primate population”. We would also like to thank the Associate Editor and the referees that helped significantly improve the manuscript by their excellent comments. We have addressed all the reviewers’ comments and suggestions in this revised version and hope that it fulfills the standard for being published in Royal Society Open Science. We have also deposited our data and codes under the following temporary Dryad doi;
<https://datadryad.org/stash/share/639EuYyb1-b5-hVvibDc5IHFzZ4bEm1SZqLThzgwSMA>.

Please, find our detailed answers to each comment below. The line numbers in our response correspond to those in the uploaded Main Document.

Sincerely,
Raisa Hernandez Pacheco

Comments to Author:

Reviewers' Comments to Author:

Reviewer: 1

This is a solid analysis and a well-written manuscript so I don't have many critical comments. It is generally acceptable for publication as is, though I do suggest (not mandate) some additions/changes that would make this manuscript perhaps better. I also have a few questions about the analysis, but these questions are simply to make sure you have robustly parameterized your matrix population model (MPM).

Thank you for the positive comments.

--It might be nice to include a life cycle graph as a figure. It took me awhile to figure out the life cycle transitions and once I did, this led me to a question/comment: The life cycle (as reflected in the projection matrix in Table 1) appears to show that a juvenile must transition to an adult female non-breeder before transitioning to failed breeder or breeder. That is, one can only go from stage 3 to stage 4, but not 3 5 or 36. Is this a biologically defensible representation of macaque fertility? Is it possible for a juvenile to transition to a failed breeder or breeder across a projection interval (year)? If females can only go from stage 3 to 4, but not stage 3 to 5 or 3 to 6, that's fine...just wanted you to double-check this.

We have included a life cycle graph as figure 2 and referenced it in the methods with further details for clarification (lines 151-165).

Given the hurricane mean matrix in Table I, we understand why readers can reach the conclusion that a juvenile must transition to an adult female non-breeder before transitioning to a failed breeder or a breeder. However, this is not correct. A juvenile can transition into any of the three adult stages (NB, FB, B), such as the mean non-hurricane matrix shows in Table I. It is true that the transitions from juveniles to breeders or failed breeders are rare and were not observed during hurricane years. Thus, we clarify this in lines (163-165). We have also re-ordered Table I to present the non-hurricane (reference) matrix first and the hurricane (treatment) matrix second.

--Another comment regarding life cycle...I expected to see three entries in the top-row of the projection matrix (the F terms in life cycle graph terminology, following Caswell 2001) corresponding to fertility (F4, F5, F6). Instead there is only 1 entry with a value of 1 at 1,6 of the projection matrix. The life cycle graph shows that one can make the following transitions 46,

56, and 66, which means that, as far as fertilities go, one would expect an entry in the 1,4 position of the projection matrix corresponding to a non-breeder who doesn't have an offspring at time t, but has an offspring at t+1. Same goes for failed breeders, the 56 transition reflects females who failed in year t, but have an offspring in year t+1, thus requiring an entry in the 1,5 position of the projection matrix. Same for the 66 transition and corresponding to the F6 entry. For example, in Brault/Caswell's 1993 paper on killer whales (in Ecology), there is a F2 entry corresponding to a juvenile who doesn't have a calf at time t, but does have one at time t+1 as it makes the transition to maternity, and there is a F3 entry corresponding to mothers who repeatedly reproduce from year to year. In any case, I just wanted to bring this to your attention--I could be wrong--so feel free to ignore if you feel you have parameterized fertility correctly.

We understand the referee's comment. It depends whether one considers the transition to occur before or after reproduction. The way we formulated it here is more common for standard matrix models and might be more intuitive than assigning fertility to a failed breeder at time t+1 because she transitioned into a breeder at t+1. The way the annual population models are currently formulated take into account the fertility of such failed breeder transitioning to a breeder at t+1 (for example) during the following annual model in which she is classified as a breeder. In this way, our model considers fertility to occur before the transition and thus NB and FB have fertility = 0. We now mention this in lines 175-176.

--In the discussion section, you could bolster some of your points by bringing in more comparative data from other primate populations. For example, in lines 294-296, you write that population growth rate is generally influenced more by survival than by fertility--this is a common finding across primates (my own review of this was in 2011 in Yearbook of Physical Anthropology) and other mammals that doesn't just pertain to climatic events: population growth rate is most sensitive to adult survival in many demographic analyses. And the difference in vital rates, stable stage distribution, reproductive value, and lambda is quite minimal between hurricane versus non-hurricane years, so bringing some comparative data along these lines might be useful to point out. More generally, the population growth rate of the Cayo population is quite high even during hurricane years, at about +10% per year and this is much higher than estimates for wild primates--perhaps some discussion of this is warranted(?).

We have included Lawler 2011 as a reference supporting our statements in lines 329-331. We have also included a statement acknowledging our study population has higher than average demographic parameters in the wild, discussed the stable stage distribution and reproductive value, and included three new comparative references (lines 321-322).

--In line 297, you mention "transient dynamics" as having a little effect. I would be careful about using the phrase "transient dynamics", as you didn't formally analyze transient dynamics, so it is not clear how you determined they have a small effect.

We agree and have deleted the sentence.

--the LTRE could be both better described in the Methods and better analyzed/discussed in the Discussion, especially for folks (e.g., psychologists, behavioral ecologists) who are not familiar with this type of analysis.

We have expanded our description of the LTRE for a broader audience in the Methods (lines 197-213), in the Results (lines 270-279), and adapted the language in the Discussion (lines 291-296).

We have also moved Table II (Table II: Life table response experiment (LTRE) showing the contributions of each stage class transition of Cayo Santiago rhesus macaque females to changes in population growth rate following hurricane years.) to the Supplementary Information and replaced it with a bar plot of the same information in the main document (figure 7). We understand a figure is a better visualization of the LTRE results as it shows better the relative (smaller) positive and (larger) negative contributions of the life cycle transitions due to hurricanes.

--In the Discussion, you might briefly mention an alternative way to incorporate environmental effects in terms of how they influence the vital rates. The present manuscript builds time-dependent MPMs for Hurricane versus non-hurricane years and then compares the (average) differences in growth rate and other vital rates across the two categories. This is fine and good, but it is also possible to model the environment directly by writing a hurricane-associated variable (rainfall, windspeed, etc.) as a covariate; in this regard, each vital rate would be a linear function of the covariate; this method is used widely (and implemented in program MARK) and also discussed by Fujiwara and Caswell in the journal Ecology (2002). Lawler et al., (2009 Oecologia) use this approach to model the effects of rainfall on vital rates in a primate population. The "covariate" approach would better allow you to actually model what you are trying to model: how hurricanes influence the vital rates. Using a MPM from a hurricane year likely subsumes all sorts of factors, both from the hurricane itself and before (e.g., feeding competition, social cohesion, dominance, etc.), and thus doesn't explicitly capture how a climatic variable directly influences a given vital rate. I'm not saying you need to do this analysis, but

you could at least mention it.

We understand there are multiple alternatives to evaluate hurricane-induced effects (e.g., covariate approach, stochasticity approach, variance, etc.). Major hurricanes are rare extreme events with potential significant consequences at the population level, but with a set of acute (short-term) environmental factors that present challenges in model parameterization and power (e.g., windspeed during 8 hours versus an annual covariate like in Lawler's approach). Following the referee's suggestion, we have included a brief last section titled "Future Directions" to mention modeling alternatives for future hurricane-related demographic studies (Lines 373-387).

--In the Discussion section, the last two sentences of the paper could be more clear. It is not clear how this analysis will provide information on "resistance or potential adaptive mechanisms opening new questions regarding the role of trade-offs between survival and reproduction...". I apologize but I'm not sure what this means? Trade-offs between survival and reproduction (in a life history sense) are set by long-term evolutionary forces and might also have clade-specific effects. (As an aside, an analysis of the trade-off between survival and reproduction in Cayo Santiago macaques was conducted by Greg Blomquist (Biology Letters, 2009), who showed that there was no phenotypic trade-off, but there was an additive genetic correlation that reflected a trade-off between longevity and reproduction). I would reword this sentence and unpack it some more to enhance the clarity. The very last sentence might be better worded as well, or at the very least one could provide references for the claim that "Our study supports evidence claiming the need" to incorporate life history traits in population studies following climatic events.

We have clarified these sentences in the Discussion (lines 366-371).

In any case, as you can see, most of my comments are basically suggestions and/or queries about the analysis. This is a nice analysis. I hope you find my suggestions helpful.

Thank you.

Reviewer: 2

Comments to the Author(s)

Review of 'Hurricane-induced demographic changes in a nonhuman primate population'
I enjoyed reading this well-written manuscript about the dynamics of an introduced rhesus macaque populations on a 15ha island of the coast of Puerto Rico. The authors ask how 3 years in which the island was hit by major hurricanes differed from the other 41 years in this impressively longitudinal demographic study. However, I also have major questions about the population models, analyses and interpretation.

First of all, I see little to no evidence that the hurricanes have affected the survival, breeding or population size of the macaques. The authors claim that there is a significant difference in projected population growth rate between non-hurricane and hurricane years ($\lambda = 1.123$ vs 1.108). But given that there were only 3 hurricane years I am not convinced that this is a

meaningful difference in lambda. Figures 2 and 5 show that there is strong overlap in the ranges of lambda values (to the degree we can talk about a range based on 3 years). The authors apply a nonparametric KS test based on the bootstrapped lambda values, but such an approach is highly problematic. P values directly depend on the number of bootstraps. Given high enough number of bootstraps, any small difference in mean lambda can be ‘proven’ significant. In addition, the bootstrapping is not stratified within years, meaning that interannual variation in environmental conditions, population size, sample size and demographic rates is ignored when comparing dynamics in non-hurricane and hurricane years.

We agree with the referee throughout the manuscript; hurricanes have a small effect on this population. Yet, our demographic data is so detailed, we can still decompose and quantify their effects (whether small or large) on the population growth rate. We have followed the referee’s suggestions and have included supplementary statistics to show a decreasing pattern in fertility during hurricane years and made changes in the language through the manuscript to be clearer on the fact that hurricanes, although small, have a role as drivers in Cayo Santiago population dynamics. We summarize our changes below.

We establish that there is a difference in mean population growth rate between years with no major hurricanes and those with major hurricanes following a bootstrap analysis on individual stage transitions (sample sizes: 18,344 for non-hurricane years and 1,816 for hurricane years). We understand the referee’s concern about the fact that our data is unbalanced and thus, we have expanded our description and justification for the use of the bootstrap in lines (180-191). Major hurricanes are rare extreme events occurring only three times in our 45-year study, but the fact that our analyses are based on data from all females (complete individual- and population-level information) and have large sample sizes makes resampling methods an appropriate approach, which are also good dealing with unbalanced data.

We also recognize there are multiple ways to test our hypotheses and that the population has substantial variation in annual population growth rate. We have added a new complementary analysis testing for differences in fertility (the vital rate with higher contribution to the decrease in population growth rate in hurricane years; see the LTRE) among treatments (non-hurricane vs hurricane years) while controlling for population density using generalized additive mixed models and model selection using Akaike’s Information Criterion. By incorporating model selection, we depart from the dependency of statistics that are highly sensitive to sample size. Yet, we are aware we can only test for correlations rather than causal relationships. We explain this new model in detail under the referee’s next comment.

We have also adopted more conservative language and have deleted the word “significant” when referring to the bootstrap results.

While we are not shown how population size fluctuates over the study period, the authors state in the Discussion that population sizes were not much different between hurricane and non-hurricane years, which could be interpreted as a sign of strong density-dependence, external population regulation and/or absence of effect of hurricanes. Without further analyses it is hard to distinguish between these factors. The authors do state that survival rates are unaffected by

hurricane years (here statistical tests would be appropriate but missing), but that there are effects on breeding probabilities. Again, these differences are not tested for statistical significance. For that purpose, and also to enable easier biological interpretation, it would be better if the authors present underlying vital rates (e.g. breeding probability conditional on survival) rather than only matrix element values. Given that only 3 hurricane years were observed and the considerable variation in lambda, I hardly expect significant differences in breeding probabilities among years caused by hurricanes.

We understand the referee's first concern about density-dependence and would like to provide a more comprehensive summary of Cayo Santiago long-term population dynamics here and in the manuscript. Previous studies from the authors and colleagues address in detail the long-term population dynamics of Cayo Santiago monkeys (see references 12,16 cited throughout the manuscript). These studies show that population dynamics is driven partly by density-dependence in fertility while no density-dependence in survival was found). Hernández-Pacheco et al (2013) tested for temporal changes in total annual survival and did not find differences across time, including during hurricane years 1989 and 1998 (now added in lines 322-323). Given these previous studies and our robust demographic analysis (also refer to our clarification of the LTRE mention in the comments of the previous referee), we are confident that survival is not significantly affected by temporal variability or by these extreme events in this population. However, we agree with the referee that population density is a major factor driving dynamics through changes in fertility, and thus it is relevant to our analysis and conclusions. Because of this, we now include a complementary generalized additive mixed model testing for differences in age-specific fertility with treatment (non-hurricane year, hurricane years) as a grouping factor and a pair of crossed random effects; the annual total number of adult females and individual ID. To include potential variability from maternal investment and maximize power, we included offspring of both sexes in this analysis (N = 12,828). Model selection using AIC indicated that treatment, although a relatively weak effect, contributes to the variation in mean age-specific fertility resulting in lower mean fertility during hurricane years. We now include the top model prediction as panel B in figure 6 (blue=non-hurricane years, red=hurricane years). Model description is now included in lines 214-229. Details of the model coefficients and selection are included as Supplementary Information.

We have deleted the sentences mentioning that population sizes were not much different between hurricane and non-hurricane years as this new analysis provide a better approach.

It is important to clarify that removal strategies (culling) are taken into account in our analysis by right censoring (now clarified in line 162-163) and thus, we can assume that culling only affects population density by lowering the number of individuals in a given year.

We do not understand the referee's comment about presenting "underlying vital rates rather than matrix element values". Matrix elements are vital rates. To avoid confusion, we use the new life cycle graph (figure 2) to clarify this in lines 172-175. The new model prediction in figure 6B can also help with this as it shows a more intuitive information.

Are there additional ways in which the authors can unveil the alleged demographic effects of hurricanes? The authors claim that hurricanes do not directly cause additional mortality, but that the most likely longer-term effect acts through defoliation. To test their hypothesis, I urge the authors to analyses annual variation in vital rates and lambda as a function of annual estimates of canopy cover. Using a continuous explanatory variable would also allow for understanding variation among years better.

Without clear patterns to show, readers are left to wonder whether or not the potential mechanisms of population response to hurricanes are relevant or not, and more importantly whether this study system is suitable for answering those questions. The population is fed regularly by humans and population sizes are regulated as well.

We discuss that defoliation and the consequent stress might be the underlying mechanisms of the effects of hurricanes. Unfortunately, we don't have information on annual canopy cover or the ability to estimate it across the 45 years of the history of the population. We do have personal communications or formal governmental reports following each hurricane reporting a major change in canopy cover and defoliation (originally cited in the paper). Given the extreme nature of these events (including the costliest event in US history), their windspeed and their trajectories through Puerto Rico and Cayo Santiago, we understand the change in landscape immediately following the hurricane we describe in the study is not in question. We want to emphasize that major hurricanes are rare extreme events with a set of acute (short-term) environmental factors that present challenges in model parameterization and power. Our study presents a first robust step towards understanding their effects in the demography of primate populations. We discuss this in the new Future Directions section (see answer to previous referee).

The constructed population models project a mean annual growth of 12%. That would mean that when starting with the original 409 monkeys, 44 years later one would count nearly 60 thousand animals. Clearly these models do not take population regulation (tetanus inoculation and removal by humans) into account. But readers have no way of assessing whether 12% growth is realistic, how much individuals are removed each year, nor how strong density dependence is in this population. This makes it hard to be confident that the constructed population models are good

representations of the population dynamics, which is important as the authors attempt to study relatively subtle effects. As an additional test, do the models project realistic life spans?

The dynamics of this population is well known since the 1970's, where no removal event nor tetanus inoculation took place in a period of a decade [Rawlins and Kessler 1986; The Cayo Santiago Macaques: History, behavior and biology. See review in our citation 18]. During this period, the annual population growth rate was reported to be 12-13%. When consequent decades of data are included in demographic models, this historical mean annual growth rate sustains (refer to Hernandez-Pacheco et al. 2013 [12]; Kessler et al. 2015; Hernandez-Pacheco et al. 2016 [16]; Hernandez-Pacheco and Steiner 2017 [33]). We explain in the methods that we follow the fate of all individuals until death or removal (right censoring, line 163). Thus, our mortality estimates are not inflated nor deflated in our analysis. Here, we built annual matrix models parameterized with observed annual vital rates (stage-specific survival and reproduction) of all females. Any effect from any mechanism affecting annual vital rates (e.g., tetanus inoculation, density) is captured in our annual matrix models. Our main motivation with this study was to decompose those effects on annual population growth rate into vital rates and test the role of major hurricanes on these changes (now explicitly mentioned in the Introduction, lines 61-63). The population grows by 12% every year on average, that is the main reason culling takes place. The robustness of these matrix models relies on the uniqueness and completeness of the demographic data used.

64-65 please explain this hypothesized trade-off in more detail

Done (lines 65-67).

77-79 please be more precise in formulating LTRE analyses

Done. Please, refer to our answer to reviewer #1.

102 why were the macaques introduced in 1938?

*We have re-worded the sentences in lines 102-105 and now they read "The Cayo Santiago Field Station (CSFS) serves as a research site for behavioral and noninvasive research on free-ranging rhesus macaques (*Macaca mulatta*). For this purpose, the rhesus macaque population was established in 1938 from 409 Indian monkeys and no other individuals have been introduced since then."*

109 why 'commercial'?

We have deleted the word "commercial" to avoid confusion. They are a free-ranging monkey colony, which means they are provisioned with food (monkey chow).

115 how were animals caught? Age-specific?

This is better explained now in the following sentences (lines 120-124): “During the trapping season some individuals have been permanently removed from the island to control for population size [16]. Annual removal strategies have varied (from no removal to up to 596 individuals removed [16, for details]) and include removal events of entire social groups, as well as age-specific and sex specific removal events. Within such structure (age and sex), individual IDs for removal are selected at random.”

119 # removed annually? Target numbers?

Removal strategies for population control have varied in an annual basis. There are years with no removal, there are years with entire social group removal, and there are other years with sex- and age-targeted removals. For details of these strategies and number of removed individuals we cite Hernández-Pacheco et al. 2016 (lines 122). We do not want to increase the length of the paper with this information as we understand it is not essential, but we can provide it per the editor’s request.

122 unclear how individuals are censused daily

Full-time staff visits Cayo Santiago from Monday to Friday to do visual censuses. We now make reference of a paper describing this process in more details (line 127).

152 I understand sons are not counted for population growth, but having sons would affect maternal investment compared to non-breeders, right?

Here, we want to address population fitness (mean annual population growth rate) which is a measure that has been proven to be robustly parameterized from female-based models. Thus, we follow general assumptions of population studies where males are assumed to not contribute to population fitness and thus their fertility is set to 0. We also don’t have complete information about paternity in this population and thus, we cannot include it in the analysis. Because of this, we based our demographic study on females. Female-based matrix models of this population in the previous studies mentioned above have proven accurate in describing population growth rate. We mention the need for two-sex models at the end of the Future Directions section.

To address the referee’s concern, we included all offspring (both sexes) in the generalized additive mixed model.

162 I do not understand this part well. So adults are classified as breeders (at time t) when they are going to have a live daughter next year ($t+1$) given that is also alive at time $t+1$? Table 1 does suggest that. But is this interpretation correct? And what does definition of ‘breeder’ mean for

the analyses of breeding probabilities and the tested trade-offs between survival and reproduction? Reproductive investment are done also when offspring do not make it to year $t+2$. Is that fully captured by the failed breeder class? My confusion also stems from a definition of breeder that apparently relies on events spread out over 2 years, while the population model has a time step of 1 year.

These females can potentially give birth every year. Thus, every year we are able to assign them stages based on their reproductive performance. Any female giving birth in any particular year (time t) will be classified as a breeder that year (time t) if her female baby survives 1 year of age (thus, the offspring survives to time $t+1$). In this way, breeders are adult females that have a baby and that baby survives 1 year of age. We have clarified and added a life cycle graph (see comments to the previous referee).

229-230 how much buffering is there compared to the net effect?

We opted to use the word “buffering” qualitatively given that the population growth rate during hurricane years changed only by $\sim 1.3\%$. We have diminished the use of the word and deleted it from the introduction.

613 I highly appreciate that R code with the constructed matrices are provided.

Thank you for the comments.